# Wheat Elongator Subunit 4 Negatively Regulates Freezing Tolerance by Regulating Ethylene Accumulation

**DOI:** 10.3390/ijms23147634

**Published:** 2022-07-11

**Authors:** Kai Wang, Mingjuan Zhai, Ran Han, Xiaolu Wang, Wenjing Xu, Xiaoxue Zeng, Guang Qi, Takao Komatsuda, Cheng Liu

**Affiliations:** 1Crop Research Institute, Shandong Academy of Agricultural Sciences/National Engineering Research Center of Wheat and Maize/Shandong Technology Innovation Center of Wheat, Jinan 252100, China; 18211084814@163.com (K.W.); hr022cn@aliyun.com (R.H.); xiaoluwang1989@hotmail.com (X.W.); wenjingxu1989@163.com (W.X.); zengxiaoxue@hotmail.com (X.Z.); guangq1102@163.com (G.Q.); takao_komatsuda@kzc.biglobe.ne.jp (T.K.); 2Biotechnology Research Institute, Chinese Academy of Agricultural Sciences, Beijing 100081, China; 15010803974@163.com

**Keywords:** Elongator complex, freezing stress, histone acetylation, ethylene, *Triticum aestivum*

## Abstract

Freezing stress is a major factor limiting production and geographical distribution of temperate crops. Elongator is a six subunit complex with histone acetyl-transferase activity and is involved in plant development and defense responses in *Arabidopsis thaliana*. However, it is unknown whether and how an elongator responds to freezing stress in plants. In this study, we found that wheat elongator subunit 4 (TaELP4) negatively regulates freezing tolerance through ethylene signaling. *TaELP4* promoter contained cold response elements and was up-regulated in freezing stress. Subcellular localization showed that TaELP4 and AtELP4 localized in the cytoplasm and nucleus. Silencing of *TaELP4* in wheat with BSMV-mediated VIGS approach significantly elevated tiller survival rate compared to control under freezing stress, but ectopic expression of *TaELP4* in *Arabidopsis* increased leaf damage and survival rate compared with Col-0. Further results showed that *TaELP4* positively regulated *ACS2* and *ACS6* transcripts, two main limiting enzymes in ethylene biosynthesis. The determination of ethylene content showed that *TaELP4* overexpression resulted in more ethylene accumulated than Col-0 under freezing stress. Epigenetic research showed that histone H3K9/14ac levels significantly increased in coding/promoter regions of *AtACS2* and *AtACS6* in *Arabidopsis*. RT-qPCR assays showed that the EIN2/EIN3/EIL1-CBFs-COR pathway was regulated by *TaELP4* under freezing stress. Taken together, our results suggest that TaELP4 negatively regulated plant responses to freezing stress via heightening histone acetylation levels of *ACS2* and *ACS6* and increasing their transcription and ethylene accumulation.

## 1. Introduction

Cold stress limits growth, reproduction, and geographical distribution of temperate crops. As for plants, cold stress is categorized into chilling stress (0–20 °C) and freezing stress (<0 °C) based on the temperatures and various physiological mechanisms that function in different temperature ranges [1,2]. Wheat (*Triticum aestivum* L.) is an important staple food crop worldwide that is fundamental in global food security. Cold stress often occurs in wheat-growing areas of the world, including China, the United States, Europe, and Australia [3,4,5,6]. Spring frost is wheat canopy temperature falling to 0 °C or below in the spring, causing severe damage to the micro-organelles of the cell, resulting in excessive reactive oxygen species (ROS) and the occurrence of lipid peroxidation [6,7,8]. At the vegetative phase, cold stress delays germination, poor emergence, and reduced plant density. Prolonged cold stress results in stunted growth, leaf chlorosis, and reduced root-shoot surface area, all of which significantly reduce wheat yield [9,10]. During the reproductive phase, cold stress damages flag leaves and spikes and reduces the number of effective tillers, grain number per spike, and grain filling rate, leading to grain yield loss [8,11,12,13].

To reduce the negative impact of cold stress to wheat yield, it is necessary to investigate cold-tolerant mechanisms and develop cold-tolerant wheat cultivars. The ICE-CBFs-COR pathway is a key pathway related to cold stress in plants. In the wheat genome, five *ICE* (inducer of CBF expression) genes, 37 *CBF* (C-repeat binding factor) genes, and 11 *COR* (cold responsive) genes have been identified [14]. TaICE41 and TaICE87 bind to the promoter of CBF gene *TaCBFIVd-B9* and activate its transcription, which confers cold tolerance on TaICE41 and TaICE87 [15]. Due to a lack of systematic research, signaling pathways, gene regulatory networks, signal transduction, and molecular mechanisms related to cold stress are poorly understood in wheat.

The Elongator complex consists of six subunits (ELP1 to ELP6) and was originally isolated as an interactor of hyperphosphorylated RNA polymerase II (RNAPII) in yeast [16,17] and later identified in humans [18] and *Arabidopsis* [19]. Structure and function studies showed that the Elongator complex is highly conserved in plants and yeast [20]. ELP1, ELP2, and ELP3 form the core subcomplex, and ELP1 and ELP2 are WD40 proteins that serve as scaffolds for complex assembly. ELP3 is the catalytic subunit harboring a C-terminal GNAT-type histone acetyltransferase (HAT) domain and an N-terminal iron-sulfur radical S-adenosylmethionine (SAM) domain [21,22,23]. ELP4, ELP5, and ELP6 form the accessory subcomplex, and each form a RecA-ATPase-like fold and assemble into a hexameric ring-shaped structure that is important for recognizing histone H3 [24,25]. Deletion of any of the six subunits results in almost identical phenotypes, suggesting that all six subunits are required for Elongator’s cellular functions [17,19,24,26,27].

In plants, elongator participates in the development and responses to biotic stresses. *Atelp* mutants exhibit various developmental defects, such as narrow and elongated leaves, short primary roots, lateral root density, abnormal inflorescence phyllotaxis, and delayed seedling growth [19,26]. A mechanistic study showed that *Atelp3* reduced histone H3K14 acetylation at the coding sequence and 3′-UTR of auxin-related genes *SHY2*/*IAA3* and *LAX2*, which decreased expression [26]. Atelp2 and Atelp3 mediate pathogen-induced transcriptome reprogramming via altering methylation levels of defense-relate genes and increasing histone acetylation levels in coding regions of several defense genes during innate immune responses [28,29]. *AtELP4* overexpression enhanced resistance to anthracnose crown rot, powdery mildew, and tomato bacterial speck in *Fragaria vesca* and tomato plants [30,31]. Previously, we isolated TaELP4 and demonstrated that it mediates immunity to *Rhizoctonia cerealis* in wheat and to *Botrytis cinerea* in *Arabidopsis* [32]. Although Elongator has been reported to regulate plant development and immunity, it is unknown whether elongator participates in cold stress.

The hormone ethylene functions directly in plant responses to freezing stress. *Arabidopsis* treated with the ethylene biosynthetic precursor ACC (1-aminocyclopropane-1-carboxylic acid) displayed decreased tolerance to freezing, but freezing tolerance was enhanced after the ethylene biosynthesis inhibitor AVG (aminoethoxyvinyl glycine) was applied. Mutants in the ethylene signaling pathway, including *etr1-1*, *ein4-1*, *ein2-5*, *ein3-1*, and *ein3 eil1*, showed enhanced freezing tolerance, while *AtEIN3*-overexpressing plants exhibited reduced freezing tolerance. Genetic and biochemical analyses revealed that ethylene negatively regulates cold signaling through AtEIN3 binding directly to the promoter of *CBF**1-3* and type-A *ARR5*, *ARR7*, and *ARR15* genes, repressing its expression [33]. Type-A *Arabidopsis* response regulators (ARRs) act as negative regulators in cold stress signaling through the inhibition of the abscisic acid–dependent pathway [34]. In soybean, the ethylene-signaling pathway inhibits the CBF/DREB1 transcripts by EIN3, contributing to the relatively poor cold tolerance of soybean [35]. Moreover, ethylene negatively regulates the tolerance of *Medicago* to freezing during cold acclimation [36].

Although many studies have elucidated cold perception and response mechanisms in plants, few have been conducted in wheat. In this study, we identified a wheat elongator subunit 4 (*TaELP4*) that had negatively regulated freezing tolerance in wheat. Functional analysis showed that *TaELP4* was induced by freezing stress; ectopic expression of *TaELP4* in *Arabidopsis* accumulates more ethylene during freezing stress by elevating histone acetylation levels in coding/promoter regions of *ACS2* and *ACS6*, and increasing their transcripts. The EIN2/EIN3/EIL1-CBFs-COR pathway was regulated by TaELP4. This work provides evidence that TaELP4 is involved in cold stress in wheat by histone acetylation and enriches research on the molecular mechanism of wheat response to cold stress. Moreover, TaELP4 might be an excellent gene that can be used to improve the freezing tolerance of wheat during breeding research.

## 2. Results

### 2.1. TaELP4 Transcripts Are Induced by Freezing Stress

To investigate whether TaELP4 participated in abiotic stress, we first analyzed the promoter regions of *TaELP4* homologs using PlantCARE (http://bioinformatics.psb.ugent.be/webtools/plantcare/html/ (accessed on 10 September 2021)). Multiple regulatory elements that respond to cold stress, drought, wounding, abscisic acid (ABA), and jasmonic acid (JA) were identified, including LTR/DRE elements (CCGAAA/GCCGAC, cold stress), MBS (CAACTG, drought-induced), WUN box (AAATTTCTT, wounding), ABRE (ACGTG, ABA-induced), and TGACG box (TGACG, JA-induced) (Figure 1A). This information revealed that TaELP4 may be involved in abiotic stress responses, such as cold responses, drought, and hormones. Sequence BLAST showed that three homologs located on chromosomes 7A, 7B, and 7D (namely *TaELP4-7A*, *TaELP4-7B*, *TaELP4-7D*) existed in Chinese Spring RefSeq v1.0 chromosomes [37]. The analysis of coding sequence (CDS) and amino acids of *TaELP4* using ClustalW software (https://www.genome.jp/tools-bin/clustalw (accessed on 1 July 2020)) showed that *TaELP4-7A*, *TaELP4-7B*, and *TaELP4-7D* had the same gene structure and high sequence similarity. Specifically, *TaELP4-7A*, *TaELP4-7B*, and *TaELP4-7D* all contained eight exons and seven introns. TaELP4-7A, TaELP4-7B, and TaELP4-7D contain a conserved PAXNEB domain (Appendix A), which is a typical domain of an RNA polymerase II Elongator protein subunit and is one part of the HAP subcomplex of elongator. The identities of CDS and amino acids ranged from 94% to 97% (Appendix A).

To investigate the biological function of *TaELP4* in detail, we first performed quantitative real-time PCR (qRT-PCR) to test the expression patterns after freezing treatment using genome-specific primers. Chinese Spring wheat was pre-treated at 4 °C for 2 h and then treated at −8 °C at the two to three leaf stage, while the control was cultured at 23 °C; the shoots were harvested at different times to test the expression pattern. Data showed that *TaELP4-7A* was significantly induced at 1 h, was highest at 2 h, and maintained the high level to 12 h. *TaELP4-7B* was induced at 0.5 h and was highest at 9 h. *TaELP4-7D* was significantly induced at 0.5 h and was highest at 3 h (Figure 1B). In general, *TaELP4* was induced by freezing stress and *TaELP4-7B* might be more essential in the wheat response to freezing stress than *TaELP4-7A* and *TaELP4-7D*.

### 2.2. Silencing of TaELP4 Improves Wheat Tolerance to Freezing Stress

Considering that three *TaELP4* homologs had high nuclear acid and amino acid identities (Appendix A), they are likely to be functionally redundant. Thus, we silenced *TaELP4* in wheat using the BSMV-mediated VIGS (virus induced gene silencing) approach to investigate the response of *TaELP4* to freezing stress. The recombinant BSMV:TaELP4 construct was generated and inoculated into seedlings at the two-leaf stage. When seedlings grew to the four-leaf stage, we randomly selected 3 BSMV:GFP and 10 BSMV:TaELP4 wheat plants to test the silenced efficiency of VIGS. The RT-qPCR (real-time quantitative PCR) results showed that *TaELP4* were all silenced and the transcripts substantially decreased 40-80% in BSMV:TaELP4-infected wheat compared to BSMV:GFP-infected plants based on primers that can simultaneously detect transcripts of *TaELP4-7A*, *TaELP4-7B*, and *TaELP4-7D* (Appendix A). This result implied that the three *TaELP4* homologs were successfully silenced.

Those plants were pre-treated at 4 °C for 2 h, then treated at −8 °C for 9 h, and subsequently recovered at 23 °C. After recovery for 2 hours, most leaves of BSMV:GFP wheat were damaged and withered by freezing. Only the first or second leaves were damaged in BSMV:TaELP4 plants, while newer leaves stayed upright (Figure 2A). After recovery for 24 h, the damaged leaves and tillers turned yellow and dry in BSMV:GFP plants, while BSMV:TaELP4 wheat grew well (Figure 2A). Tiller survival rate showed 80% surviving tillers in BSMV:GFP plants, significantly lower than BSMV:TaELP4 wheat, which was nearly 100% (Figure 2B). To further study the influence of freezing on tillers, we kept the freeze-treated plants in the greenhouse. One month later, almost all BSMV:GFP plants died, and the tiller survival rate decreased to 10%, while BSMV:TaELP4 wheat grew well with normal jointing, with a tiller survival rate of 58% (Appendix A). These results imply that freezing stress severely damaged wheat growth and *TaELP4*-silenced wheat was more tolerant to freezing stress than control.

### 2.3. Ectopic-Expression of TaELP4 Decreased Tolerance to Freezing Stress in Arabidopsis

To investigate whether *TaELP4* overexpressing lines were more sensitive to freezing stress, we over-expressed *TaELP4-7B* in *Arabidopsis thaliana*. Two representative *TaELP4*-overexpressing transgenic lines (OE-1 and OE-2) in the T_3_ generation were selected in this study. We first tested the expression of TaELP4 in *Arabidopsis* using confocal microscopy. GFP fluorescence signal showed that TaELP4 localized at the cytoplasm and nucleus (Appendix A). The Col-0 and transgenic lines were pre-treated at 4 °C for 2 h, then treated with −8 °C for 2 h, and finally recovered at 22 °C. After recovery for 2 h, there was no difference between TaELP4-overexpressing lines and Col-0 (Figure 3A). After recovery for 5 days, some plants withered and died, and OE-1 and OE-2 had more leaves with severe freezing injury than Col-0 (Figure 3B). Statistical analysis showed that plant survival rates of OE-1 and OE-2 were 52% and 59%, while Col-0 had a survival rate of 70% (Figure 3C). This demonstrated that *TaELP4* overexpression decreased freezing tolerance in *Arabidopsis*.

Subcellular localization showed that TaELP4 localized at both the cytoplasm and nucleus in *Arabidopsis*, so we wanted to confirm that this localization was correct. We cloned *TaELP4* and *AtELP4* genes and separately constructed them into p16318h-GFP vector, then transformed them into wheat mesophyll protoplast cells. Subcellular localization assay showed that both TaELP4 and AtELP4 localized at the cytoplasm and nucleus and co-localized with a nuclear marker OsMADS3 (Appendix A). We then extracted the total proteins to detect whether TaELP4 and AtELP4 were intact. Western blot showed that TaELP4-GFP and AtELP4-GFP were intact proteins with molecular weights of 68 kDa (Appendix A). This result implied that TaELP4 might have the same function as AtELP4, and TaELP4 had normal function in *Arabidopsis*.

### 2.4. TaELP4 Regulated Ethylene Biosynthesis

The hormone ethylene negatively regulates tolerance to freezing in *Arabidopsis*, soybean, and *Medicago* [33,35,36]. To investigate whether TaELP4 negatively regulates freezing-tolerance through the ethylene pathway, we tested the expression of *ACS2* and *ACS6*, two key rate-limiting enzymes that catalyze the committing reaction in ethylene biosynthesis [38]. First, we tested the transcript levels of *TaACS2* and *TaACS6* in *TaELP4*-silenced and BSMV:GFP-infected plants after freezing treatment for 3 h. The RT-qPCR results showed that the transcripts of *TaACS2* and *TaACS6* were significantly decreased in *TaELP4*-silenced plants compared to BSMV:GFP plants (Figure 4A), suggesting that TaELP4 might regulate ethylene biosynthesis by regulating the transcripts of *TaACS2* and *TaACS6*. We then tested the transcripts of *AtACS2* and *AtACS6* in OE-1 and OE-2 as well as in Col-0 after freezing treatment for 2 h. The RT-qPCR results showed that the transcripts of *AtACS2* and *AtACS6* were increased in OE-1 and OE-2 lines compared with Col-0 (Figure 4B), suggesting that TaELP4 positively regulates the expression of *AtACS2* and *AtACS6* in *Arabidopsis*.

The result that TaELP4 positively regulated *AtACS2* and *AtACS6* gene transcription in *Arabidopsis* prompted us to test the production of ethylene in OE-1, OE-2, and Col-0. We examined the ethylene production in transgenic plants and Col-0 plants after freezing treatment. We found that ethylene production in Col-0 was 637.8 nl/g/h, while the values in OE-1 and OE-2 were 1181.5 and 2067.8 nl/g/h, respectively (Figure 4C), which were 1.85 and 3.24-fold higher than Col-0. This result showed that *TaELP4* transgenic plants accumulated high levels of ethylene in *Arabidopsis* during freezing stress.

### 2.5. TaELP4 Increases Histone H3K9/14ac Levels of AtACS2 and AtACS6 in Arabidopsis

Previous papers reported that the acetylation level of histone H3 is generally associated with active transcription and TaELP4 epigenetically regulates immune responses to *R. cerealis* and *B. cinerea* in wheat and *Arabidopsis* through facilitating chromatin histone H3K9/14 acetylation of defense genes [32,39,40]. To test whether TaELP4 regulates histone acetylation levels in ACS2 and ACS6, ChIP assay via histone H3K9/14ac (Histone H3 acetylated at lysines 9 and 14) antibody was deployed to examine histone H3 acetylation level in regions of *AtACS2* and *AtACS6* in OE-1 and OE-2 as well as in Col-0. ChIP-qPCR results showed that histone H3K9/14ac levels in the promoter and coding regions of *AtACS2* and *AtACS6* were significantly higher in OE-1 and OE-2 compared to Col-0 (Figure 5). These data indicated that *TaELP4* overexpression boosted histone H3K9/14ac levels in the promoter and coding regions of *AtACS2* and *AtACS6*, which were consistent with higher transcript levels of *AtACS2* and *AtACS6* in *TaELP4*-transgenic plants than in Col-0.

### 2.6. TaELP4 Regulates the EIN3/EIL1-CBFs-CORs Pathway

In *Arabidopsis*, ethylene negatively regulates cold signaling partially through direct transcriptional control of cold-regulated CBFs-COR genes by EIN3 [33]. We tested whether TaELP4 regulated the EIN3/EIL1-CBFs-CORs pathway during responses to freezing stress. RT-qPCR showed that *AtEIN2*, *AtEIN3*, and *AtEIL1* had no differences between *TaELP4*-overexpressing lines and Col-0 under normal conditions. After freezing stress, *AtEIN2*, *AtEIN3*, and *AtEIL1* transcripts in Col-0 were decreased, and the transcripts were significantly higher in OE-1 and OE-2 than in Col-0, implying that TaELP4 boosted *AtEIN2*, *AtEIN3*, and *AtEIL1* transcription under freezing stress. We next tested the expression levels of *AtCBF1*, *AtCBF2*, and *AtCBF3*, and showed that *AtCBF2* and *AtCBF3* were down-regulated in OE-1 and OE-2 compared with Col-0 under normal conditions and freezing stress, while *AtCBF1* showed no differences under normal conditions and was significantly lower in OE-1 and OE-2 than in Col-0 under freezing stress. This demonstrated that TaELP4 down regulated *AtCBFs* expression. In *Arabidopsis*, three CBFs (CBF1, CBF2, and CBF3) bind to the promoters of COR genes and then activate expression in a large subset of COR genes under cold stress, including *AtRD29*, *AtCOR15b*, *AtCOR47*, and *AtKIN1* [33,41,42]. We further detected the expression levels of selected COR genes and showed that *AtRD29* and *AtCOR47* were down regulated in OE-1 and OE-2 compared with Col-0 during freezing stress, while *AtCOR15b* and *AtKIN1* showed no difference, which implied that TaELP4 regulated *AtRD29* and *AtCOR47* expression. Moreover, we found that TaELP4 overexpression decreased type-A ARR genes *AtARR5* under freezing stress (Figure 6). Those results showed that TaELP4 regulated the EIN3/EIL1-CBFs-COR pathway to respond to freezing.

## 3. Discussion

Cold stress often occurs in the growth season of crops, including wheat and rice [2,10]. The genetic and molecular mechanisms of plant responses to cold stress have been identified in rice and *Arabidopsis*, but are much less known in wheat. In this study, we found that TaELP4 was a negative regulator in wheat responses to cold stress. Several LTR/DRE elements that responded to cold stress existed in the promoter of *TaELP4*, and the expression of *TaELP4* homologs were induced by freezing at different times (Figure 1), implying that TaELP4 participates in freezing stress. Further evidence showed that TaELP4 is a negative regulator of freezing tolerance. When we silenced three *TaELP4* homologs simultaneously in wheat, BSMV:TaELP4 wheat significantly improved resistance to freezing stress compared to BSMV:GFP (Figure 2 and Appendix A). When we overexpressed *TaELP4* in *Arabidopsis*, *TaELP4* transgenic lines showed severe injury and a higher death rate than Col-0 after freezing treatment (Figure 3). Those results further showed that TaELP4 negatively regulates freezing tolerance. We know that the soil is a buffer preventing rapid effects of temperature, however, in this freezing experiment, we can see the *Arabidopsis* that lived at the edge of the pot that died first; we speculated that −8 °C may cause the soil to become frozen and that may have caused severe damage to plants with underdeveloped roots. Once thawed, the roots cannot quickly recover to meet normal growth, resulting in the death of plants. Previous papers documented that AtELP1-6 complex regulates plants growth, development, and immunity [19]. In wheat, TaELP4, acting as a functional regulator, is required for immune the response of wheat to *R. cerealis* infection as with the response of *Arabidopsis* to *B. cinerea* [32]. Thus, we deduced that ELP4 has multiple functions.

Ethylene is a versatile phytohormone, and ethylene-signaling participates in cold stress [43]. In *Arabidopsis*, ethylene-insensitive mutants, including *ein2* and *ein3 eil1,* displayed enhanced tolerance to freezing stress by increasing the expression of cold-inducible CBF1/2/3 and type-A ARR genes [33]. Cold stress induces *CBFs* expression, which can activate expression in a series of COR genes [41,42]. Here, we found that TaELP4 positively regulated *ACS2* and *ACS6* expression, two key rate-limiting enzymes that catalyze the committing reaction in ethylene biosynthesis, and elevated ethylene accumulation after freezing stress (Figure 4). These results implied that TaELP4 might regulate freezing tolerance through ethylene signaling. Next, we tested the expression of ethylene signaling transduction components and downstream genes and showed that *TaELP4* overexpression increased the expression of *AtEIN2*, *AtEIN3*, and *AtEIL1*, and reduced the expression of *AtCBF1*, *AtCBF2*, *ACBF3*, *AtRD29*, *AtCOR47*, and *AtARR5* (Figure 6). These results demonstrated that *TaELP4* overexpression elevated ethylene content and regulated the EIN2/EIN3/EIL1-CBFs-COR pathway after freezing stress.

Elongator complex was originally identified in yeast as a histone acetyltransferase complex that activates RNAPII [17]. The acetylation level of histone H3 is generally associated with active transcription [28,29,39,40]. A recent study showed that TaELP4 up-regulates transcription of a subset of defense-related genes through elevating the histone H3K9/14ac levels of their chromatin, in turn boosting plant innate immune responses [32]. In this study, we demonstrated that *TaELP4* overexpression in *Arabidopsis* elevated the histone H3K9/14ac levels of *AtACS2* and *AtACS6* in the promoter and gene body region (Figure 5). This result implies that *TaELP4* overexpression increases H3K9/14ac levels of *AtACS2* and *AtACS6,* and then elevates their expression during freezing stress. This result links TaELP4 overexpression to enhanced ethylene content during freezing. However, how TaELP4 directly affects histone acetylation level is not clear, and whether there are expression or sequence differences of TaELP4 between winter wheat and spring wheat is not clear, and needs to be studied in a further study.

## 4. Materials and Methods

### 4.1. Plant Materials

Chinese Spring wheat (*Triticum aestivum* L.) was used to investigate *TaELP4* transcription profiles and in VIGS experiments. *Arabidopsis thaliana* Columbia-0 (Col-0) ecotype was used to ectopically-express *TaELP4-7B*.

### 4.2. Freezing Tolerance Assay and Phenotype Analyses

Wheat plants were grown in a greenhouse at 23 °C/14 h light (250–300 μmol m^−2^ s^−1^) and 10 °C/10 h dark. Seedlings at the four-leaf stage were pre-treated at 4 °C for 2 h, then treated at −8 °C for 9 h, and subsequently recovered at 23 °C. The wheat growth state was photographed and the tiller survival rate was calculated. The tiller survival rate (%) was calculated as: 100% × dead tiller number/total tiller number. *Arabidopsis* were grown in a greenhouse at 22 °C/14 h light (100–150 μmol m^−2^ s^−1^) and 22 °C/10 h dark. Plants before flowering (about 20 days after germination) were cultured at 4 °C for 2 h, then treated at −8 °C for 2 h, and subsequently recovered at 22 °C. After recovery for 5 days, the plant survival rate (%) was calculated as 100% × dead plant number/total plant number. The plants with all dead rosette leaves were defined as death after recovery for 5 days. The freezing experiments were repeated at least twice and processed in the dark.

### 4.3. RNA Extraction and RT-qPCR

Total RNA was extracted from wheat or *Arabidopsis* samples using Trizol reagent (Invitrogen, Carlsbad, CA, USA). RNA was first purified and then checked for integrity. For RT-qPCR, total RNA was reverse-transcribed to cDNA using the FastQuant RT Kit (Tiangen, Beijing, China). RT-qPCR was carried out in a Roche LightCycler480 system using SYBR Premix Ex Taq kit (TaKaRa, Shiga, Japan). Reactions were set up using the following thermal cycling profile: 95 °C for 30 s followed by 40 cycles of 95 °C for 5 s, 58 °C for 30 s, and 72 °C for 34 s. Each experiment was replicated three times. The relative expression of the target genes was calculated using the 2^−ΔΔCT^ method [44], with the wheat Actin gene (*TaActin*) or *Arabidopsis* actin gene (*AtActin*) used as the internal reference genes for the analysis.

### 4.4. Plasmid Construction and Plant Transformation

To generate BSMV-TaELP4 vector for barley stripe mosaic virus (BSMV)-mediated VIGS experiment, a 272 bp fragment matching the CDS of *TaELP4-7A*, *TaELP4-7B*, and *TaELP4-7D* (from 414 to 686 nucleotides in *TaELP4-7B* sequence) was selected based on SiFi software and then subcloned in antisense orientation into the *Nhe* I restriction site of the RNA γ of BSMV. The inoculation method followed the protocols described by Holzberg et al. [45] and Wang et al. [32].

To generate the *35S:TaELP4-7B-GFP* vector for ectopic expression, the CDS of *TaELP4-7B* was amplified and cloned into pCAMBIA1300-GFP (*Hind* III) using in-fusion PCR cloning systems. The resulting *35S:TaELP4-7B-GFP* vector was introduced into *Arabidopsis* Col-0 plants by *Agrobacterium*-mediated transformation [46]. Transgenic plants were selected by hygromycin B, and T3 homozygous transgenic plants were used in this study.

To generate *35S:TaELP4-7B-GFP*, *35S:AtELP4-GFP* vector for subcellular localization and the CDS of *TaELP4-7B* or *AtELP4* were respectively amplified and cloned into p16318h-GFP (*BamH* I) using in-fusion PCR cloning systems.

### 4.5. Subcellular Localization and Western Blot

To detect the subcellular localization of TaELP4 and AtELP4 proteins, *35S:TaELP4-7B-GFP* and *35S:AtELP4-GFP* vectors were transformed into wheat protoplasts via PEG-calcium transfection method, following the protocol of Yoo et al. [47]. Nuclear marker OsMADS3 was used as the co-localization marker. GFP fluorescence was observed by confocal microscopy (ZEISS710; Carl Zeiss, Jena, Germany) with excitation light at 488 nm.

For Western blot, the transformed wheat protoplasts were collected by centrifugation at 100× *g* for 2 min, and then total proteins were extracted using plant protein extraction kit (CW0885M, CWBIO, Taizhou, China). Total soluble proteins (~20 μg) were separated on 12% SDS-PAGE and transferred to polyvinyl difluoride membranes (Amersham). The blotting membranes were incubated with 2500-fold diluted Anti-GFP Mouse Monoclonal Antibody (TransGen Biotech, Beijing, China) at 4 °C overnight and then incubated with 4000-fold diluted Goat Anti-Mouse IgG (H + L), HPR conjugated secondary antibody (TransGen Biotech, China) at 22–23 °C for 1 h. The fusion protein was visualized using the Pro-light HRP Chemiluminescent Kit (TransGen Biotech, China).

### 4.6. Measurement of Ethylene Content

Ethylene emission was measured with a gas chromatograph as described previously [48]. In brief, seedlings (20–25) of Col-0 and *TaELP4*-overexpression transgenic plants were grown in 15 mL gas chromatography vials containing 5 mL 1/2 Murashige & Skoog medium. Seedlings were cultured under a 16 h: 8 h, light: dark cycle at 22 °C for 2 weeks. Seedlings were pre-treated at 4 °C for 2 h, and then treated at −8 °C for 2 h, and finally the vials were sealed for 12 h at 22 °C. In total, 1 mL of gas from each vial was used to analyze the ethylene emissions with a gas chromatograph (Hitachi, Tokyo, Japan).

### 4.7. Chromatin Immunoprecipitation with Acetylated-Histone3 Lysine 9/14 and qPCR

Chromatin immunoprecipitation (ChIP) was performed as described by Gendrel et al. [49]. Briefly, approximately 2.0 g of *Arabidopsis* plants after freezing treatment were cross-linked in 1% formaldehyde under a vacuum. Chromatin was extracted and fragmented via ultrasound treatment to a size of 200–500 bp. The fragments were incubated with acetylated Histone H3 K9/K14ac (Histone H3 acetylated at lysines 9 and 14) antibody (Santa Cruz Biotechnology, Heidelberg, Germany), and the immune complex was collected by Dynabeads protein G (Invitrogen). After extensive washing, the immunoprecipitated chromatin was recovered using the phenol-chloroform method. The amount of precipitated DNA corresponding to a specific gene region was determined by qPCR and normalized by both input DNA and a constitutively expressed gene *AtActin*.

### 4.8. Statistical Analyses

All experimental results are means ± standard deviations of at least three independent replicates. Student’s *t*-test was used for comparing two data sets in the SPSS System.

### 4.9. Accession Numbers

Sequence data from this article can be found in the EnsemblPlants (http://plants.ensembl.org/index.html (accessed on 2 June 2021)) or TAIR (https://www.arabidopsis.org/ (accessed on 2 June 2021)) databases under accession numbers: TaELP4-7A (TraesCS7A02G522900), TaELP4-7B (TraesCS7D02G512100), TaELP4-7D (TraesCS7B02G439900), TaACS2 (TraesCS2B02G414800), TaACS6 (TraesCS2A02G396400), TaActin (TraesCS5D02G516200), AtACS2 (AT1G01480), AtACS6 (AT4G11280), AtEIN2 (At5g03280), AtEIN3 (At3g20770), AtEIL1 (At2g27050), AtCBF1 (At4g25490), AtCBF2 (At4g25470), AtCBF3 (At4g25480), AtRD29 (At5g52310), AtCOR15b (At2g42530), AtCOR47 (At1g20440), AtKIN1 (At5g15960), AtARR5 (At3g48100), and AtActin (At3g18780).

## 5. Conclusions

TaELP4 negatively regulates freezing tolerance by regulating ethylene signaling. *TaELP4* was induced by freezing stress, which elevates the histone H3K9/14ac levels of *AtACS2* and *AtACS6*, and increases the expression of *AtACS2* and *AtACS6*, resulting in more ethylene content being accumulated. Ethylene signaling negatively regulates freezing tolerance through CBFs-COR pathway (Figure 7). This study provides insights on how TaELP4 negatively regulates freezing tolerance in monocot plant species.

## Figures and Tables

**Figure 1 ijms-23-07634-f001:**
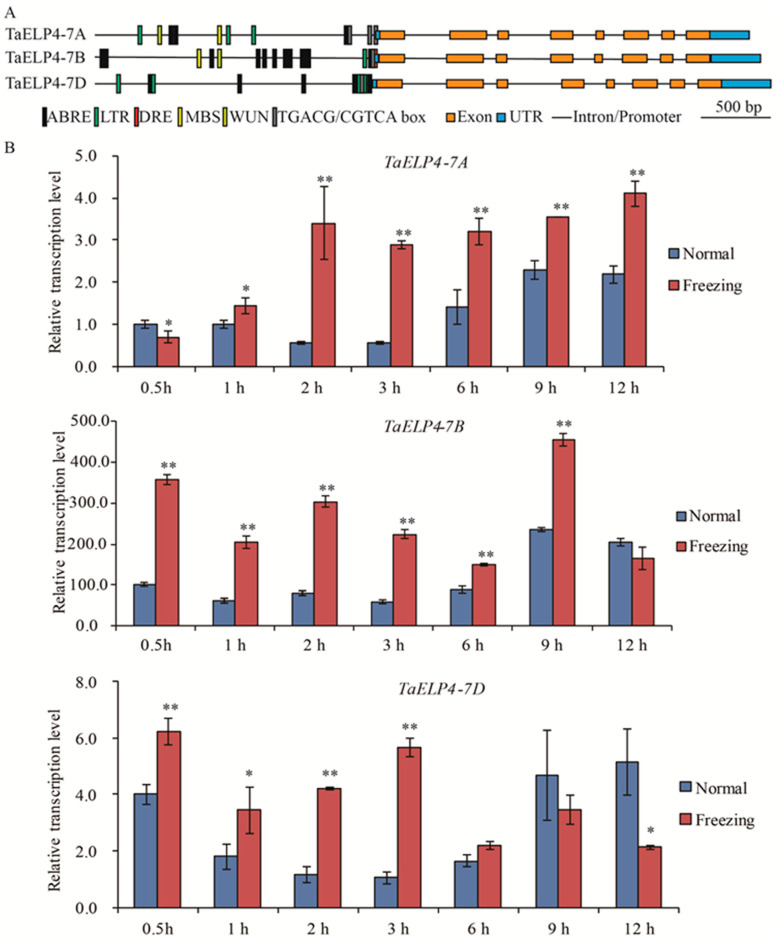
Gene structure, promoter cis-acting element, and expression pattern analysis of *TaELP4* homologs. (**A**) The gene structure of TaELP4 homologs, including 2000 bp region upstream of the ATG start codon in the promoters, untranslated region, intron, and exon. Promoter cis-elements analyzed using PLACE (http://bioinformatics.psb.ugent.be/webtools/plantcare/html/ (accessed on 10 September 2021)). (**B**) The relative transcript levels of *TaELP4-7A*, *TaELP4-7B*, and *TaELP4-7D* in shoots of Chinese Spring at 0.5, 1, 2, 3, 6, 9, and 12 h with −8 °C treatment (Freezing) or 23 °C (Normal). The transcript level of *TaELP4-7A* at 0.5 h of normal condition was set to 1. Asterisks (*, *p* < 0.05; **, *p* < 0.01) indicate significant differences between normal and freezing based on Student’s *t*-test. Bars indicate the standard deviation of the mean; *TaActin* was the internal control for normalization of gene expression.

**Figure 2 ijms-23-07634-f002:**
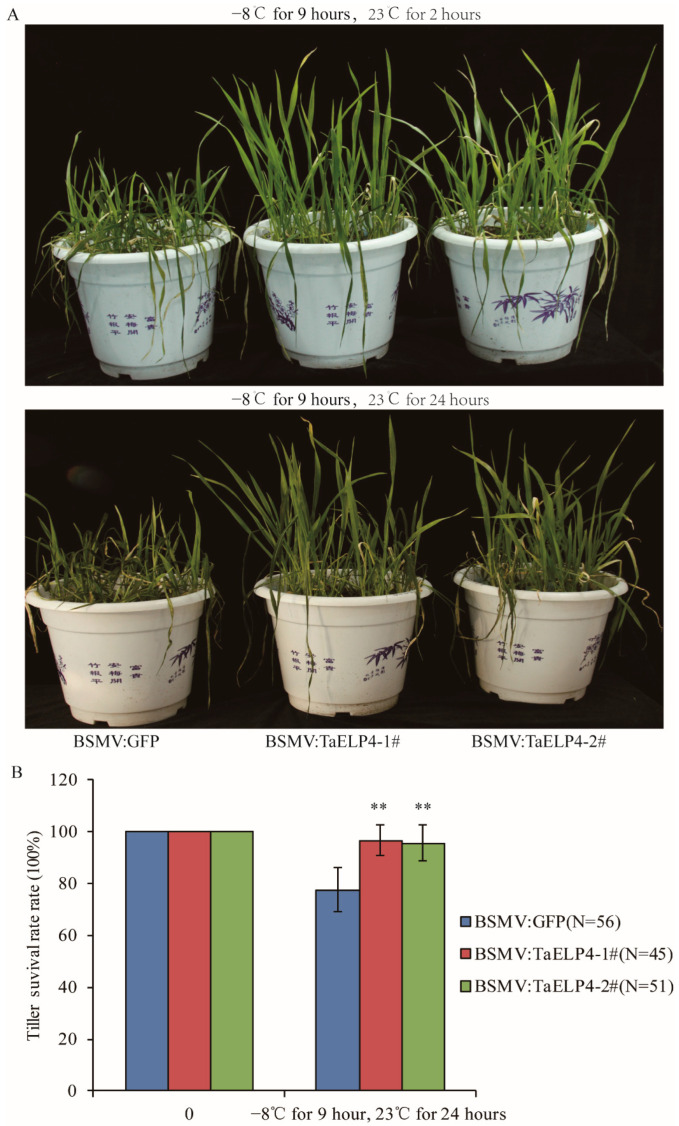
Phenotypes and tiller survival rates of BSMV-TaELP4 and BSMV-GFP with freezing stress. (**A**) BSMV-TaELP4 and BSMV-GFP wheat seedlings at the four-leaf stage were pre-treated at 4 °C for 2 h, then treated at −8 °C for 9 h, and subsequently recovered at 23 °C for 2 and 24 h; the phenotypes are shown. (**B**) The surviving tillers from the heart leaf without freezing injury were calculated. N means numbers of tillers. Double asterisks (**, *p* < 0.01) indicate significant differences between BSMV-TaELP4 and BSMV-GFP wheat based on Student’s *t*-test. Bars indicate the standard deviation of the mean.

**Figure 3 ijms-23-07634-f003:**
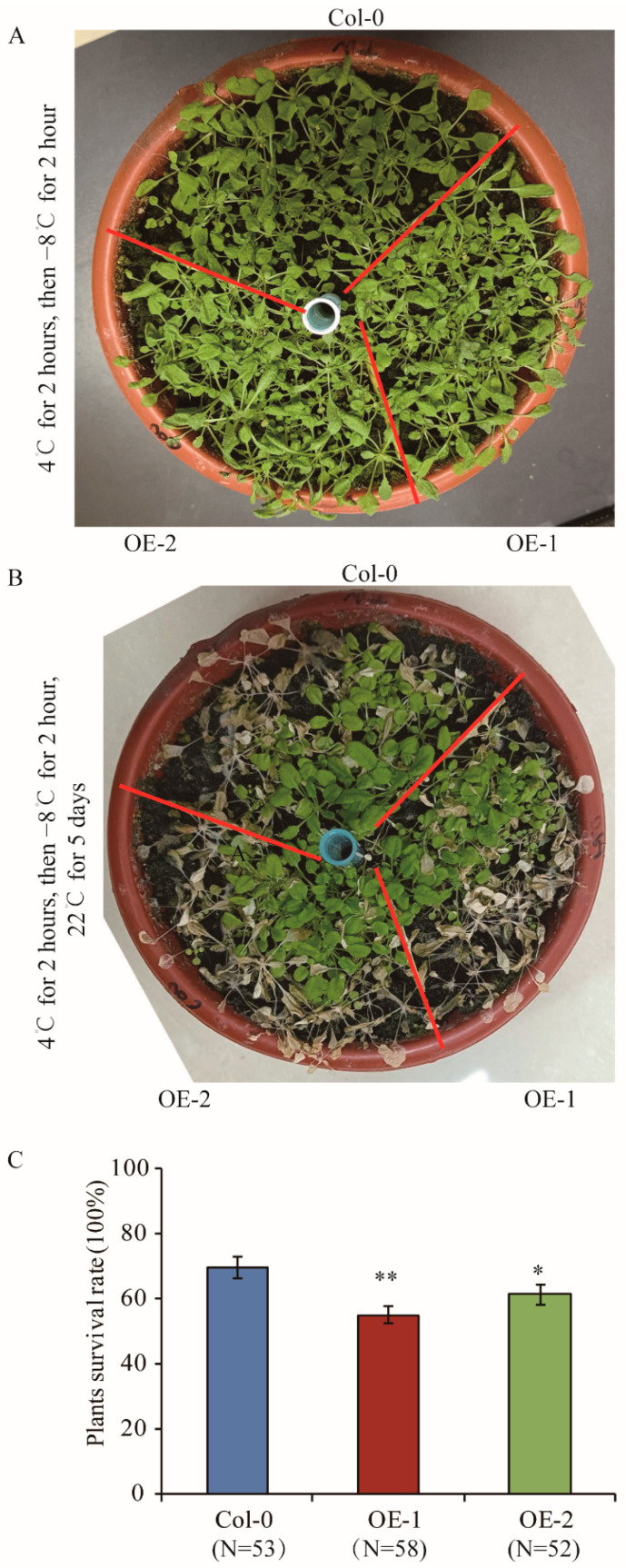
Phenotypes and plant survival rates of *TaELP4*-overexpression transgenic *Arabidopsis* and Col-0 with freezing stress. (**A**,**B**) Plants of *TaELP4*-overexpression transgenic *Arabidopsis* and Col-0 before flowering were pre-treated at 4 °C for 2 h, then treated at −8 °C for 2 h, and subsequently recovered at 22 °C for 2 h and 5 days; the phenotypes are shown. (**C**) The surviving plants without withering were calculated. N means numbers of plants. Asterisks (*, *p* < 0.05; **, *p* < 0.01) indicate significant differences between *TaELP4*-overexpression transgenic *Arabidopsis* and Col-0 based on Student’s *t*-test. Bars indicate the standard deviation of the mean.

**Figure 4 ijms-23-07634-f004:**
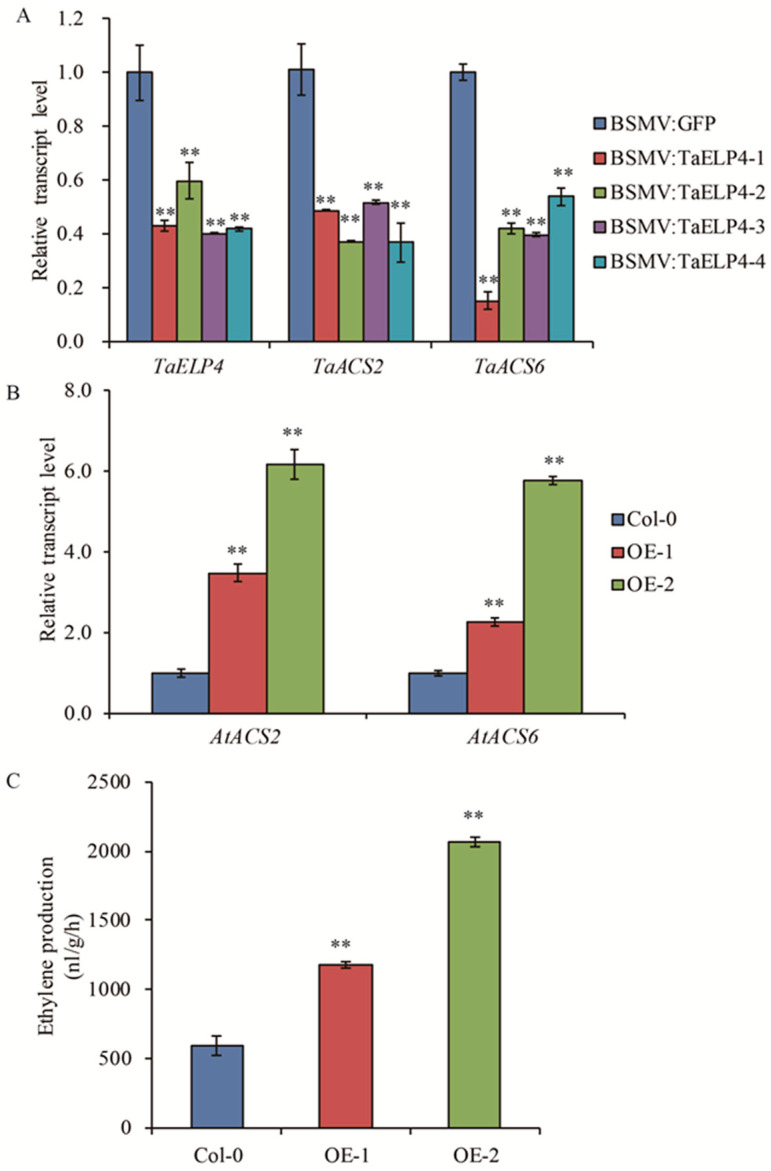
TaELP4 positively regulates expression of *ACS2* and *ACS6* and enhances ethylene content. (**A**,**B**) The relative transcript levels of *ACS2* and *ACS6* in *TaELP4*-silenced and -overexpressed plants and their control after freezing treatment. *TaActin* or *AtActin* was the internal controls for normalization of gene expression. (**C**) Ethylene production in *TaELP4*-overexpressed and Col-0 after freezing treatment. Double asterisks (**, *p* < 0.01) indicate significant differences based on Student’s *t*-test. Bars indicate the standard deviation of the mean.

**Figure 5 ijms-23-07634-f005:**
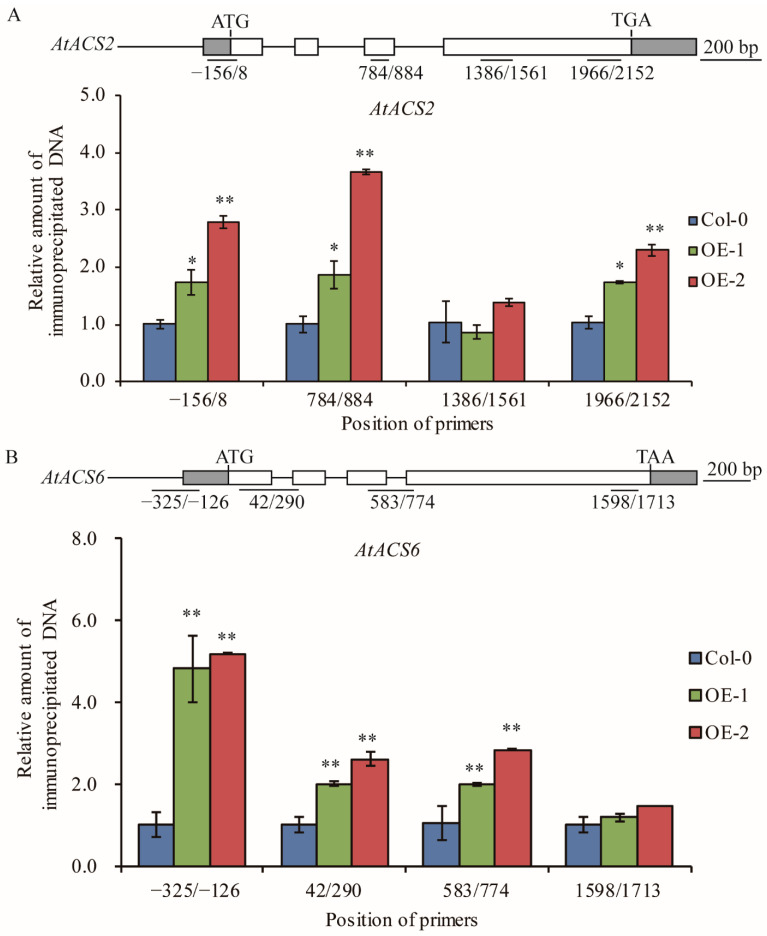
Histone H3K9/14 acetylation levels of *AtACS2* (**A**) and *AtACS6* (**B**) in *TaELP4*-overexpressed transgenic *Arabidopsis* and Col-0. The position of the primers is relative to the initiation ATG codon. The relative amount of immunoprecipitated chromatin fragments (as determined by qPCR) from *TaELP4*-overexpressing lines were compared with that from Col-0. *AtActin* was the internal control. Asterisks (*, *p* < 0.05; **, *p* < 0.01) indicate significant differences based on Student’s *t*-test. Bars indicate standard error of the mean.

**Figure 6 ijms-23-07634-f006:**
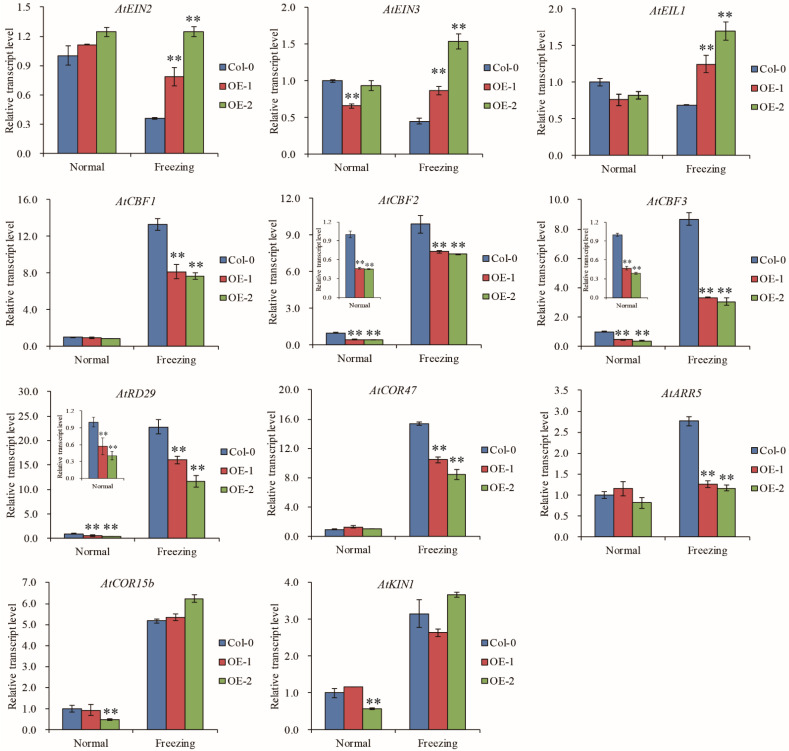
The relative expression levels of ethylene-signaling components, CBFs, and selected COR genes in *TaELP4*-overexpressing lines and Col-0. The relative expression levels of *AtEIN2*, *AtEIN3*, *AtEIL1*, *AtCBF1*, *AtCBF2*, *AtCBF3*, *AtRD29*, *AtCOR47*, *AtARR5*, *AtCOR15b*, and *AtKIN1* in *TaELP4*-overexpressing lines and Col-0 with or without freezing treatment. The transcription level in Col-0 was set to 1, *AtActin* was the internal control for normalization of gene expression. Asterisks (**, *p* < 0.01) indicate significant differences based on Student’s *t*-test. Bars indicate the standard deviation of the mean.

**Figure 7 ijms-23-07634-f007:**
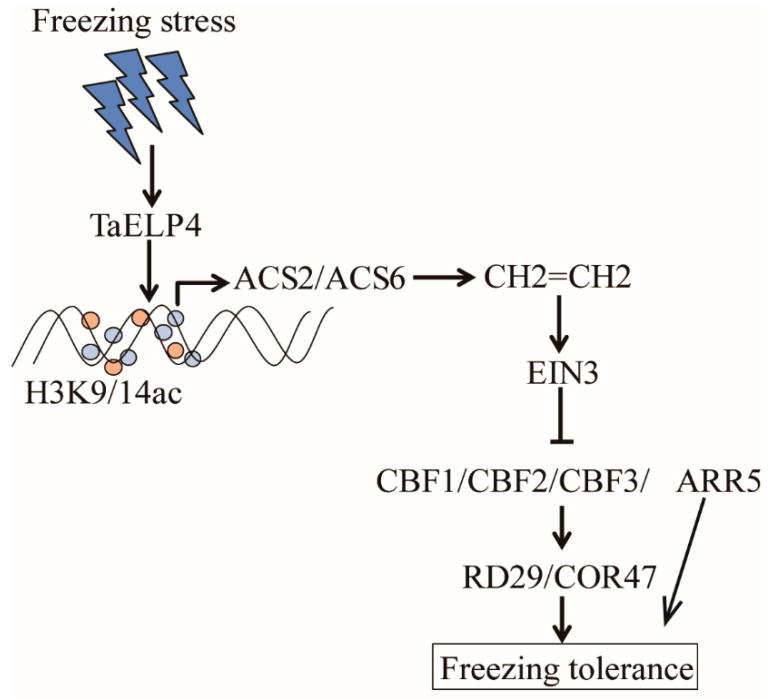
Proposed model of TaELP4 negatively regulates freezing tolerance. Freezing stress induces *TaELP4* expression, which elevates the histone H3K9/14ac levels of *AtACS2* and *AtACS6*, and increases expression of *AtACS2* and *AtACS6*, resulting in more ethylene content accumulated. Ethylene signaling negatively regulates CBFs-COR pathway to reduce freezing tolerance.

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
