# Peer review of "Wheat Elongator Subunit 4 Negatively Regulates Freezing Tolerance by Regulating Ethylene Accumulation"

_ijms, 2022, doi:10.3390/ijms23147634_

Round 1

Reviewer 1 Report

The manuscript by Kai Wang & Mingjuan Zhai et al. is an interesting contribution to our knowledge of cold and freezing tolerance in crop plants. There are several drawbacks and some amendable errors, but I believe that these can be addressed in a revision. 

Minor issues:

General

- There are some typos/style inconsistencies (e.g., missing a preposition at L64, a font size inconsistency at L234). 

Introduction

- The type-A ARR genes are generally known as cytokinin-responsive genes, the role in cold stress was found only for a subset of ARRs. Include corresponding references or provide a more detailed description to clarify. 

Methods

- The authors repeatedly stated that seedlings were used for the experiment. That is not correct (a 20-day-old plant is not a seedling).

- Light quality and intensity are important factors in cold response, yet none of that is commented on, and information about light quality/photon-flux density is completely missing. That must be provided. 

- The protoplast preparation step is not described in the methods.

- 'http://202.194.139.32/blast/blast.html' is an incorrect citation for the WheatOmics portal

- Please indicate the exact process for determining freezing stress / cold acclimation. The soil is a buffer preventing rapid effects of temperature, and it is thus unlikely that more than the leaf tissue was exposed to the stated temperature for the given period. This should be at least discussed in the manuscript.

- Survival assay. In my experience, five days of recovery does not seem to be sufficient for Arabidopsis. Please, indicate the parameter that was used for determining survival. 

Results

- L241 - The authors claim that TaELP4-7A was significantly induced at 1 h. However, that seems to be less than 1.5-fold and on the threshold for the sensitivity of standard qPCR. Please, indicate the fold-change threshold used for determining significant differences. 

- Indicate how many biological and technical replicates were used for all presented data, and provide a more detailed description, i.e., how many plants were used per biological replicate?

- The authors concluded that TaELP4-7D is less important for freezing stress response. However, the quantitative comparison of the three isoforms is missing. What is the estimated proportion of ELP expression? 

- The experiment presented in Figure 1 should be supplemented with information about the diurnal rhythm. It seems that the presented data were collected during the light period. That should be indicated and commented on. Note that multiple bars are missing standard deviation.  

- The COR genes expression profile. These are only examples of COR genes, and that should be indicated (e.g., add 'selected COR genes')

In conclusion, I believe that once these comments are addressed, the manuscript will be suitable for IJMS.

Reviewer 2 Report

Dear Editors,

Thank you so much for choosing me as a reviewer of the manuscript ijms-1792281 entitled: ,,Wheat Elongator subunit 4 regulates tolerance to freezing stress by regulating ethylene accumulation”. I hope that my comments will help Authors to improve their manuscript.

Detailed remarks concerning the manuscript:

According to the guidelines for Authors the title of your manuscript should be concise, specific and relevant. It should identify if the study reports (human or animal) trial data, or is a systematic review, meta-analysis or replication study. Please do needed changes.

According to the guidelines for Authors in the Abstract question addressed in a broad context and highlight the purpose of the study should be placed please do needed changes.

According to the guidelines for Authors the “Introduction” section should define the purpose of the work and its significance, including specific hypotheses being tested. Please do needed changes. Besides the clear answer for the question stated as specific scientific hypothesis hypotheses being tested should be given

According to the guidelines for Authors in the Abstract the main methods or treatments applied should be briefly described. Please do needed changes.

Key words. It is not recommended to use as a key words the words or phrases that appeared in the title of the manuscript. Please do needed changes.

Please cite the references in the text of the manuscript according to the scheme presented in the guidelines for Authors. See: “In the text, reference numbers should be placed in square brackets [ ], and placed before the punctuation; for example [1], [1–3] or [1,3]. For embedded citations in the text with pagination, use both parentheses and brackets to indicate the reference number and page numbers; for example [5] (p. 10). or [6] (pp. 101–105).” Please do needed changes.

Please present Latin name of the Chinese Spring wheat used to investigate TaELP4 transcription profiles and in 117 VIGS (virus induced gene silencing) experiments.

All the figures should be clear for the reader without referring to the text of the manuscript. Please do the appropriate changes where needed.

Some of the sentences in the “Results” section sounds like the discussion and includes references citation. See subsection 3.5. TaELP4 increases histone acetylation of AtACS2 and AtACS6 genes in Arabidopsis

Conclusions should be somewhat expanded. Please propose the direction for the future studies. Please avoid figure citation in the conclusions.

References in the reference list should be presented according to one scheme presented in the guidelines for Authors. There are many editorial mistakes in the reference list. It is impossible to mention all of them. There are some examples. Once the abbreviated journal titles, but the other time full titles of journals are presented. The names of the Authors of cited manuscripts are incorrectly presented, Does they should be bolded? Also the way (scheme) of the names and surnames of the Authors for cited references are wrong. See: Journal Articels: Author 1, A.B.; Author 2, C.D. Title of the article. Abbreviated Journal Name YearVolume, page range; Books and book chapters:  Author 1, A.; Author 2, B. Book Title, 3rd ed.; Publisher: Publisher Location, Country, Year; pp. 154–196.3. Author 1, A.; Author 2, B. Title of the chapter. In Book Title, 2nd ed.; Editor 1, A., Editor 2, B., Eds.; Publisher: Publisher Location, Country, Year; Volume 3, pp. 154–196. The year of publication should be bolded and presented in the another place Once all words in the title of the manuscript in the reference list are written with capital letter, but the other time only first word in the title of the manuscript are written with capital letter. Please do needed changes
